# Electrocardiographic Characteristics, Identification, and Management of Frequent Premature Ventricular Contractions

**DOI:** 10.3390/diagnostics13193094

**Published:** 2023-09-29

**Authors:** Dimitris Tsiachris, Michail Botis, Ioannis Doundoulakis, Lamprini Iro Bartsioka, Panagiotis Tsioufis, Athanasios Kordalis, Christos-Konstantinos Antoniou, Konstantinos Tsioufis, Konstantinos A. Gatzoulis

**Affiliations:** 1First Department of Cardiology, School of Medicine, National and Kapodistrian University of Athens, “Hippokration” Hospital, 11527 Athens, Greece; mgmpotis94@gmail.com (M.B.); doudougiannis@gmail.com (I.D.); ibartsioka@gmail.com (L.I.B.); ptsioufis@gmail.com (P.T.); akordalis@gmail.com (A.K.); ckantoniou@hotmail.gr (C.-K.A.); ktsioufis@gmail.com (K.T.); kgatzoul@med.uoa.gr (K.A.G.); 2Athens Heart Center, Athens Medical Center, 15125 Athens, Greece

**Keywords:** premature ventricular contractions, catheter ablation, electrocardiography, management algorithm

## Abstract

Premature ventricular complexes (PVCs) are frequently encountered in clinical practice. The association of PVCs with adverse cardiovascular outcomes is well established in the context of structural heart disease, yet not so much in the absence of structural heart disease. However, cardiac magnetic resonance (CMR) seems to contribute prognostically in the latter subgroup. PVC-induced myocardial dysfunction refers to the impairment of ventricular function due to PVCs and is mostly associated with a PVC burden > 10%. Surface 12-lead ECG has long been used to localize the anatomic site of origin and multiple algorithms have been developed to differentiate between right ventricular and left ventricular outflow tract (RVOT and LVOT, respectively) origin. Novel algorithms include alternative ECG lead configurations and, lately, sophisticated artificial intelligence methods have been utilized to determine the origins of outflow tract arrhythmias. The decision to therapeutically address PVCs should be made upon the presence of symptoms or the development of PVC-induced myocardial dysfunction. Therapeutic modalities include pharmacological therapy (I-C antiarrhythmic drugs and beta blockers), as well as catheter ablation, which has demonstrated superior efficacy and safety.

## 1. Epidemiology

Premature ventricular complexes (PVCs) constitute one of the most common arrhythmias encountered in clinical practice. Contemporary data suggest an increasing incidence of idiopathic PVCs, defined as PVCs without apparent structural heart disease [1]. The prevalence of PVCs is proportional to the monitoring duration. A study among 122 043 US Air Force personnel demonstrated at least 1 PVC in 8 per 1000 participants, during a 48-s ECG recording [2]. PVCs were detected in 5.5% of participants in the Atherosclerosis Risk in Communities (ARIC) study, based on a 2 min ECG [3].

Notably, there are no epidemiological data on the prevalence high burden (>5 or 10%) idiopathic (i.e., without concomitant structural heart disease) PVCs in random Holter monitoring data. When 24 h Holter monitoring was implemented among healthy adults, 69% of participants had at least one PVC, with a median PVC count of 2 and a 95th percentile of 193 PVCs [4]. In a population of 1424 community-dwelling individuals who underwent Holter monitoring, increasing age, smoking status, elevated systolic blood pressure, and impaired left ventricular ejection fraction were independently associated with increased PVC frequency [5]. Even though most patients with PVCs experience no symptoms, PVC-related symptoms can lead to worsened quality-of-life and include palpitations, chest discomfort and dizziness [6]. Interestingly, periods of bigeminy and trigeminy appear most commonly during rest and cause pseudo-bradycardia or, more accurately, bradysphygmia and subsequent fatigue since PVCs often do not generate peripheral pulse and impair cardiac output.

## 2. Natural History

Regarding asymptomatic idiopathic PVCs natural history, the existing body of literature provides discordant findings. Day-by-day fluctuations in PVC burden have been reported in patients undergoing 14-day monitoring, but most data are based on 24 h recordings [7].

Gaita et al. [8] reported that among 55 patients who were followed up for 15 years, PVCs spontaneously resolved in half of them. Similarly, among a paediatric population without structural heart disease, PVCs resolved during follow-up in 28% of patients (estimated mean time to disappearance 115 months) [9]. Lee et al. more recently reported spontaneous remission in 44% of study population, at a median follow-up time of 15.4 months [10]. Gatzoulis et al. reported similar rates of satisfactory response between antiarrhythmic drug recipients (82%) and non-recipients (86%), among idiopathic RVOT symptomatic patients [11].

In contrast, no significant change in PVC prevalence was reported in 239 asymptomatic patients with normal left ventricle, over 5.6 years [12]. It should be noted that this study included patients with an initially diagnosed high PVC burden of 10%. Initiation point of follow-up is crucial since increased PVC burden is usually asymptomatic and remission, if ever occurring, this will take place in the first months.

## 3. Prognosis

PVCs have consistently been associated with adverse cardiovascular outcomes among patients with pre-existing structural heart disease. The association between PVCs and increased mortality in the context of ischemic heart disease is well established, as numerous historical studies provide relevant data [13,14,15]. Initial efforts for pharmacological suppression of PVCs in survivors of myocardial infarction, using class IC antiarrhythmics, led to detrimental effects on total mortality [16], now known to stem from the untoward interplay between abnormal substrate (scar and ischemia) and pharmacodynamics of this particular class. Concerning non-ischemic systolic heart failure, a sub-group analysis of the DANISH study, among patients with left ventricle ejection fraction (LVEF) ≤ 35%, high PVC burden (defined as more than 30 PVCs per hour) was associated with a 38% higher risk of death from any cause and a 78% higher risk of cardiovascular mortality [17].

The association between the presence of PVCs in patients with apparently normal hearts and adverse cardiovascular outcomes has been a matter of debate. A hallmark study in 1985 reported no difference in prognosis between 70 asymptomatic, healthy participants with ventricular ectopy and the general population [18]. In the first meta-analysis of prospective cohort studies of participants without clinically apparent heart disease, the presence of PVCs was associated with worse cardiovascular outcomes [19]. However, the exclusion of structural heart disease was not based on advanced tests, and the risk was also correlated with presence of cardiovascular risk factors.

The Cardiovascular Health Study (CHS) was a landmark study of 1429 24 h ambulatory ECG recipients that demonstrated a 31% increased risk of death and a 48% increased risk for incident heart failure in the upper quartile of PVC frequency compared to those in the lowest quartile [20].

In the context of resolving this discrepancy, cardiac magnetic resonance (CMR) has emerged as a pivotal tool in the assessment of PVCs prognosis. In a cohort of 518 patients in whom PVCs were deemed as asymptomatic after negative routine diagnostic workup, subjects with myocardial abnormalities on CMR were at increased risk for the composite outcome of sudden cardiac death, resuscitated cardiac arrest and episodes of ventricular fibrillation-ventricular tachycardia [21]. Patients with multifocal PVCs and non-left bundle branch block morphology had increased incidence of abnormal CMR findings.

## 4. Arrhythmia Mechanism and Substrate

In the modern era, an absence of structural heart disease has to be confirmed by CMR. In a single-center prospective study of patients with significant PVC burden, (>1.000 but <10.000, or presence of NSVT) and normal findings at echocardiography, structural heart disease was diagnosed in 25.5% (mostly myocarditis) and non-specific abnormal findings were present in 20% [22].

The mechanism of idiopathic frequent PVCs is not related to myocardial scarring, suggesting either an increased triggered activity or the presence of a micro-reentry [23,24]—also explaining the tendency to flare up during lower heart rates. The sites of successful catheter ablation of idiopathic PVC origins have been progressively elucidated and include both the endocardium and, less commonly, the epicardium. Idiopathic PVCs usually originate from specific anatomical structures such as the ventricular outflow tracts, aortic root, atrioventricular annuli, papillary muscles, Purkinje network, and exhibit characteristic electrocardiograms based on their anatomical background [25].

With respect to PVCs originating specifically from the right ventricular outflow tract, subclinical myocarditis not detectable with current CMR techniques seems to be the underlying mechanism. In the case of aortic root, extensive fibrous tissue presents between the base the cusps and LV myocardium appears as the most plausible factor [26].

## 5. PVC-Induced Myocardial Dysfunction

An increasing body of evidence has revealed that frequent PVCs can lead to impairment of ventricular function (PVC-induced cardiomyopathy), which is a potentially reversible condition after PVC elimination [27]. PVC burden has been shown to be the strongest independent predictor of PVC-induced cardiomyopathy in several studies [27,28,29].

The cut-off value that separates those at high risk of PVC-induced cardiomyopathy ranges from 16% to 26%, in different studies [29,30,31]. Current guidelines report that a PVC burden of 10% seems to be the minimal threshold for development of LV dysfunction, with higher risk when the PVC burden is >20% [27,29].

Other risk factors include PVC-QRS width of >150 ms [32], asymptomatic status [33], and interpolated PVCs [34]. Male sex and epicardial origin of the arrhythmia are also independent predictors of PVC-induced myocardial dysfunction [35]. Elimination of PVCs often leads to improvement or resolution of cardiomyopathy [29,31]. Interestingly, therapeutic benefit of PVC suppression extends to patients with concomitant, but causally unrelated, structural heart disease [36]. A wider QRS during SR is a poor prognostic marker for EF recovery following PVC catheter ablation [37].

## 6. Prevalence of Outflow Tract Ventricular Arrhythmias

The most common PVC site of origin remains a matter of debate, as the epidemiological data regarding frequency of various PVC types derive mainly from tertiary healthcare datasets, implying selection bias. However, PVCs from the outflow tract (OT) appear to be the most common [38]. The majority of outflow tract ventricular arrhythmias (OTVAs) originate from the right ventricular outflow tract (RVOT) (70–80%) [39]. Less frequently (15–25%), the site of origin is the left ventricular outflow tract (LVOT) [40].

## 7. Electrocardiographic Patterns of RVOT Sites

Anatomically, the RVOT should be perceived as a tubular structure that wraps around the LVOT, such that it is always located anterior and (counter-intuitively) leftward relative to the LVOT [41]. It is separated into the rightward (also referred to as free wall), anterior, leftward, and posterior (also referred as septal) portions. The posterior RVOT is anterior to the LVOT, and its portion is adjacent to the right coronary cusp (RCC) and part of the left coronary cusp (LCC).

Pace mapping in the RVOT region manifests site-specific ECG morphologies. Dixit et al. demonstrated that septal sites produced taller and narrower R waves in inferior leads, compared with free wall sites [42]. A possible explanation is the sequential activation of the ventricles during free wall stimulation.

Further distinctive features were described by Jadonath et al. Pacing at an anterior septal site produced a predominant QS morphology (either QS or Qr) in lead I, whereas pacing posteriorly produced consistently either an R or an Rs complex. Lead aVL was also of discriminatory value, as pacing at the anterior septal sites always produced a QS patter, whereas posterior septal placing always demonstrated an R wave within the QRS complex (qR, rS, rSr morphology) [43].

## 8. Electrocardiographic Patterns of Pulmonary Valve Sites

Histopathological studies demonstrate the presence of ventricular myocardial extensions into the pulmonary artery, beyond the ventriculoarterial junction, in a relatively common fashion (17% in autopsy-derived data) [44], which may serve as arrhythmogenic substrate. The prevalence of pulmonary artery originated ventricular arrhythmias is 4%, among patients referred for idiopathic outflow tract VT/PVC ablation [45]. Ventricular arrhythmias arising from the pulmonary artery exhibit taller R waves on inferior leads and a greater aVL/aVR Q-wave amplitude ratio, compared with RVOT arrhythmias [46]. These findings are compatible with the more leftward-anterior location of the pulmonary valve.

Anatomically, the cusps of the pulmonary valve are defined according to their relationship with the aortic valve and thus are termed anterior (or nonseptal), right and left cusp. The three cusps are not situated at the same level: the left cusp is located at the lowest level, with anterior and right cusps being located relatively superior. Liao et al. reported smaller R wave amplitude in the inferior leads and greater QRS duration in ventricular arrhythmias arising from the right cusp, compared with the anterior and left cusp [47].

## 9. Electrocardiographic Patterns of LVOT Sites

Approximately one third of idiopathic VA originates from the LVOT [48]. The aortic valve is centrally situated and exhibits anatomical proximity to each of the remaining cardiac valves [41]. The anatomical ventriculoarterial junction (VAJ) serves as the boundary between the LV part of the myocardial septum and the fibroelastic wall of the aortic trunk [49]. The base of the aortic valve sinuses contains ventricular musculature, as the hinge lines of the aortic cusps extend below the VAJ. These muscular structures, along with the commissures, are the ablation sites for LVOT arrhythmogenicity, as opposed to the coronary cusps, commonly mentioned in the relevant literature [50]. Access to the myocardial tissue at the bases of the aortic valve sinuses is readily obtained transaortically, retrogradely crossing into the left ventricular outflow tract, while a transeptal approach may also suffice, if the ventricular tissue above leaflet hinges is of limited extent [49].

Pacing the left coronary aortic sinus (LCAS) produces a multiphasic QRS morphology, with an M- or W-shaped pattern in lead V1, and a transition zone no later than V2 [51]. In contrast, arrhythmias arising from the more anterior right coronary aortic sinus (RCAS) produce a later transition zone, typically at lead V3 [52]. Non-coronary sinus rarely produces ventricular arrhythmias and pacing from this region typically results in atrial capture [51].

Arrhythmias from the RCAS/LCAS commissure exhibit a qs morphology in lead V1 with notching on the downward deflection and a transition zone in lead V3 [53]. Yamada et al. reported a distinct qrS morphology in leads V1–V3 when pacing this site [54].

## 10. Electrocardiographic Patterns of LV Summit

The LV summit was first described by McAlpine in 1975 and it constitutes the most superior aspect of the epicardial LV [55]. It is surrounded by the left anterior descending artery, the left circumflex artery, and the arc between the first septal perforating branch and the left circumflex artery. The great cardiac vein (GCV) bisects the LV summit into a lateral area, which is accessible to catheter ablation, and a central area, which is inaccessible to catheter ablation, due to the anatomic proximity with the coronary arteries [56].

The accessible area demonstrates typical right bundle branch block (RBBB) morphology. As the site of origin shifts from the great cardiac vein towards the anterior intraventricular vein, the ECG pattern transforms into left bundle branch block (LBBB) morphology, with reversal of R-wave transition in V1 [57]. A characteristic ECG morphology is the precordial “pattern break”, defined as an abrupt loss of the R wave in lead V2, compared with leads V1 and V3, which is indicative of an origin adjacent to the anterior intraventricular sulcus and is associated with poor ablation outcomes [58].

## 11. ECG Prediction Algorithms

### 11.1. Classic ECG Algorithms (Table 1 and Table 2)

#### 11.1.1. Earliest Onset in V2 and Time to the First QRS Peak/Nadir in Leads V2 and III

Yang et al. [59] compared the electrocardiographic characteristics between PVCs or VT originating from the RVOT and the aortic sinus cusps in 45 patients. The authors established that if any of the following criteria is fulfilled, the more likely site of origin is the aortic sinus cusps. First, lead V2 does not have the earliest QRS onset. Second, the time from the earliest QRS onset among the 12 leads to initial QRS peak/nadir in lead II is ≥120 msec. Third, the time from the earliest QRS onset to initial QRS peak/nadir in lead V2 is ≥78 msec. The sensitivity and specificity were calculated 92% and 88%, respectively. The earliest onset in V2 during RVOT arrhythmias is likely explained by the closer anatomic location.

#### 11.1.2. R Wave Duration Index-R/S Amplitude Index

Zhang et al. [60] used R wave duration index, defined as R-wave/QRS duration index of the PVC in lead V1 or V2, to differentiate RVOT from LVOT origin. A cutoff less than 0.5 was predictive of RVOT-PVC. Moreover, R/S wave amplitude index, calculated as R-wave/S wave amplitude of the PVC in lead V1 or V2 less than 0.3 was supportive of RVOT origin. The combined sensitivity of the two measures was 94.87, with a 100% positive predictive value. Additionally, the authors reported a transitional zone ≥ V4 as 92.3% sensitive for RVOT-PVC (Figure 1).

#### 11.1.3. V2 Transition Ratio

Betensky et al. [61] introduced a two-step approach to differentiate ventricular arrhythmia sites, among patients with precordial transition zone at lead V3. A PVC precordial transition later than SR transition had 100% specificity for RVOT origin. If the PVC transition occurred at or earlier compared with SR transition, the V2 transition ratio, defined as PVC R-wave/QRS amplitude, divided by SR R-wave/QRS amplitude, was calculated. A ratio ≥ 0.6 was suggestive of LVOT origin with a sensitivity of 95% and a specificity of 100% (Figure 2).

#### 11.1.4. Transitional Zone Index

Yoshida et al. [62] defined the transitional zone (TZ) index as TZ score of outflow tract ventricular arrhythmia, minus the TZ score of sinus rhythm. The TZ score was determined according to the precordial lead where R/S transition occurred; if the R/S amplitude ratio was greater than 0.9 and less than 1.1, the TZ score was same as the lead number. If the TZ was observed between two leads, 0.5 was incremented to the earlier lead (e.g., in a transition zone between V3 and V4, the TZ score was calculated 3.5). A transitional zone index of <0 differentiated ASC origin from RVOT origin, with 88% sensitivity and 82% specificity (Figure 3).

#### 11.1.5. V3 R-Wave Deflection Interval Combined with V1 R-Wave Amplitude Index

Further diagnostic measures to distinguish idiopathic ventricular arrhythmias with transitional lead at V3 were developed in that algorithm [63]. The R-wave deflection interval, defined as the site of earliest ventricular activation in lead V3 to the peak of the R-wave < 80 msec, combined with R-wave amplitude index in lead V1 < 0.30, predicted RVOT origin with a sensitivity of 100% and a specificity of 83.3% in the prospective validation cohort (Figure 4).

#### 11.1.6. V2S/V3R Amplitude Index

The V2S/V3R amplitude index [64], defined as the ratio of the S wave amplitude in lead V2 to the R wave amplitude in lead V3 during the ventricular arrhythmia, was evaluated among 207 patients with an LBBB pattern and inferior axis outflow tract ventricular arrhythmias. A cut-off value of ≤1.5 separated LVOT origin from RVOT origin with 89% sensitivity and 94% specificity. The smaller ratio in LVOT ventricular arrhythmias is presumably determined by the higher R/S wave amplitude ratio in precordial leads, when arrhythmias originate from the more posterior LVOT structures [64] (Figure 5).

#### 11.1.7. V1–V2 S-R Amplitude Difference

Kaypakli et al. [65] introduced the S-R amplitude difference in lead V1 through V2, defined as (V1S + V2S) − (V1R + V2R). A cut-off value of >1.625 predicted RVOT origin with 95.1% sensitivity and 85.5% specificity. The anatomic proximity of RVOT with leads V1–V2, compared with LVOT-originated arrhythmias, leads to increased S wave amplitude and deceased R wave amplitude, which explains these findings [65] (Figure 6).

#### 11.1.8. Combined Transition Zone and V2S/V3R Amplitude Index

He et al. [66] established a novel mathematical formula, Y = −1.15 X (Transition Zone Index) − 0.494 × (V2S/V3R), based on a retrospective comparison of previously described ECG algorithms. A cut-off value ≥ 0.76 predicted an LVOT origin with 90% sensitivity and 87% specificity in the prospective validation cohort. The predictive accuracy of the mathematical formula was higher, compared with six previously described algorithms.

#### 11.1.9. Lead I R-Wave Amplitude

Xie et al. [67] reported that PVC R wave amplitude in lead I greater than 0.1 mV is predictive of LVOT arrhythmia origin. This was tested in a retrospective cohort of 75 patients, with 75% specificity and 98% sensitivity. The authors attributed their findings to the anterior and leftward location of the RVOT, relative to the LVOT.

#### 11.1.10. Initial R Wave Surface Area Index (ISA)

Nikoo et al. [68] developed the Initial R Wave Surface Area Index (ISA), which is derived by multiplying the R wave duration, measured in milliseconds, with the R wave amplitude, measured in mV, in lead V1 or V2, during PVC or VT. A value greater than 15 could distinguish LVOT from RVOT arrhythmias with 78.2% sensitivity and 94.6% specificity, in a retrospective cohort of sixty patients (Figure 7).

#### 11.1.11. V1–V3 Transition Index

Further distinctive features between RVOT and LVOT arrhythmias were described by Di et al. [69]. The V1–V3 transition index is defined as the sum of the ratios of S wave amplitude in lead V1 and V2 during ventricular arrhythmia to the S wave amplitude during sinus rhythm, decremented by the sum of the ratios of R wave amplitude during ventricular arrythmia from lead V1 to lead V3 to the R wave amplitude during sinus rhythm (i.e., [SPVC/SSR(V1) + SPVC/SSR(V2)] − [RPVC/RSR(V1) + RPVC/RSR(V2) + RPVC/RSR(V3)]). The index produced higher values for RVOT sites compared with LVOT sites, with a threshold of −1.60 demonstrating 95% accuracy for RVOT sites and 75% accuracy for LVOT sites in a prospective validation cohort (Figure 8).

#### 11.1.12. RV1-V3 Transition Ratio

Recently, Efremidis et al. [70] devised the RV1-V3 transition ratio, which is calculated by dividing the sum of R wave amplitude from lead V1 to V3, during PVC, to the sum of R wave amplitude from lead V1 to V3, during sinus rhythm [i.e., (RV1PVC + RV2PVC + RV3PVC)/(RV1SR + RV2SR + RV3SR)]. The authors included patients with idiopathic outflow tract PVCs and precordial transition in lead V3. The ratio was lower for RVOT-originating arrhythmias, with 94% sensitivity and 73% specificity for a threshold of 0.9 (Figure 9).

#### 11.1.13. R-S Difference Index

Zhao et al. [71] introduced the R-S Difference Index, to differentiate RVOT septum from LVOT-ASC PVCs. The R-S Difference index is defined as the sum of R wave amplitudes from lead V2 to V4, minus the S wave amplitude of lead V1, during PVCs [i.e., RPVC(V2) + RPVC(V3) + RPVC(V4) − SPVC(V1)]. An R-S Difference Index greater than 20.9 was indicative of LVOT origin, with 73.7% sensitivity and 86.3% specificity. The authors attributed their findings to the anterior anatomical location of RVOT compared with LVOT-ASC, which leads to smaller R-wave amplitudes of the precordial leads and larger S-wave amplitude of lead V1 (Figure 10).

**Table 1 diagnostics-13-03094-t001:** Conventional lead algorithms—methodology.

Author	Year Published	Site Differentiation	Patients Included	Study Methodology	Electroanatomical Mapping	Inclusion Criteria
Yang et al. [59]	2007	RVOT vs. ASC	45	Retrospective	31 patients	Symptomatic-Refractory VT Antiarrhythmic drugs refractoriness Absence of structural heart disease
Zhang et al. [60]	2009	RVOT vs. LVOT	52	Retrospertive cohort: 39 patients Prospective cohort: 13 patients	Yes	Monomorphic VA with LBBB morphology and inferior axis. Normal LVEF Failed or intolerant beta-blocker/antiarrhythmic therapy
Betensky et al. [61]	2011	RVOT vs. LVOT	61	Retrospertive cohort: 40 patients Prospective cohort: 21 patients	Yes	Idiopathic PVC or VT Precordial transition in lead V3 Patients with presumed cardiomyopathy due to frequent ventricular ectopy were included
Yoshida et al. [62]	2011	RVOT vs. ASC	112	Retrospective	NR	Symptomatic idiopathic VT or PVCs, successfully ablated in either RVOT or ASC ECG with typical LBBB morphology with inferior axis Normal ECG during SR
Cheng et al. [63]	2012	RVOT vs. LVOT	43	Retrospertive cohort: 31 patients Prospective cohort: 12 patients	NR	At least one failed antiarrhythmic drug treatment Precordial transition in lead V3
Yoshida et al. [64]	2014	RVOT vs. LVOT	207	Retrospective	Yes	Absence of structural heart disease LBBB morphology–inferior axis of VT/PVC
Kaypakli et al. [65]	2017	RVOT vs. LVOT	123	Retrospective	22 patients	Symptomatic patients Frequent outflow tract PVCs Successful ablation Not fulfilling ARVC task force criteria
He et al. [66]	2018	RVOT vs. LVOT	695	Retrospertive cohort: 488 patients Prospective cohort: 207 patients	YES	Successful outflow tract ventricular arrhythmias with LBBB and inferior axis Absence of structural heart disease
Xie et al. [67]	2018	RVOT vs. LVOT	75	Retrospective	YES	Ventricular arrhythmias with LBBB morphology and inferior axis Absence of structural heart hisease
Nikoo et al. [68]	2020	RVOT vs. LVOT	60	Retrospective	NR	Symptomatic VT or PVC, refractory to pharmaceutical therapy, with inferior axis Absence of structural heart disease Successful ablation
Di et al. [69]	2019	RVOT vs. LVOT	184	Retrospertive cohort: 147 patients Prospective cohort: 37 patients	YES	Symptomatic outflow tract VT or PVC Precordial transition in lead V3 Absence of ischemic-structural heart disease and paced rhythm
Efremidis et al. [70]	2021	RVOT vs. LVOT	58		YES	Outflow tract ventricular VT or PVC with LBBB and inferior axis Precordial transition in lead V3 Absence of structural heart disease
Zhao et al. [71]	2022	Septum of RVOT vs. LVOT-ASC	259	Retrospective	NR	Idiopathic PVCs Successful ablation

**Table 2 diagnostics-13-03094-t002:** Conventional lead algorithms—diagnostic measures.

Author	Algorithm	Diagnostic Measures
Yang et al. [59]	V2 is not the lead with the earliest QRS onset, OR Time to initial QRS peak/nadir in lead III > 120 msec, OR Time to initial QRS peak/nadir in lead V2 > 78 ms: Predicts ASC origin.	Sensitivity 92% Specificity 88%
Zhang et al. [60]	Transition zone ≥ V4 predicts RVOT origin R wave duration index [QRS duration (PVC)/R duration (PVC)] in V1 or V2 < 0.5 OR R(PVC)/S(PVC) amplitude index < 0.3 in V1 or V2 predicts RVOT origin	Precision 100% Sensitivity 92.30% for Criterion A Precision 100% Sensitivity 94.87% for Criterion B
Betensky et al. [61]	PVC precordial transition later than SR transition predicts RVOT origin, IF NOT V2 Transition Ratio: VT (R/R + S) divided by SR(R/R + S) ≥ 0.60 predicts LVOT origin	Specificity 100% Sensitivity 19% for Criterion A Precision 100%, Specificity 100% Sensitivity 95%, AUC 0.992 for Criterion B
Yoshida et al. [62]	Transition Zone index, defined as Transition Zone score (calculated according to the position where the amplitudes of the R and S waves are equal) of OT-VA minus Transition Zone score of sinus beat. TZ index < 0 predicts ASC origin	Sensitivity 88% Specificity 82% AUC 0.9
Cheng et al. [63]	R wave deflection interval (ascending part of R in PVC) in V3 > 80 msec predicts LVOT VT, IF NOT R wave amplitude index (R wave amplitude divided by QRS amplitude during PVC) in V1 > 0.3 predicts LVOT VT	Precision 85.70% Sensitivity 100% Specificity 83.30%
Yoshida et al. [64]	V2S/V3R (in PVC) ≤ 1.5 predicts LVOT origin	Precision 84.0% Sensitivity 89.0% Specificity 94.0% AUC 0.964
Kaypakli et al. [65]	S-R difference: (V1S + V2S) − (V1R + V2R) > 1.625 mV predicts RVOT origin.	Precision 86.50% Sensitivity 95.10% Specificity 85.0% AUC 0.929
He et al. [66]	Y = −1,15 × (TZ) − 0.494 × (V2S/V3R). Y ≥ −0.76 predicts LVOT origin.	Sensitivity 90.0% Specificity 87.0% AUC 0.88
Xie et al. [67]	R wave amplitude ≥ 0.1 in I predicts LVOT origin	Precision 92.30% Sensitivity 98.0% Specificity 75.0% AUC 0.85
Nikoo et al. [68]	ISA index: multiplying R wave in msec by the R wave amplitude in mV in the leads V1 or V2. A cut off value ≥ 15 in any of these leads predicts LVOT origin.	Precision 94.60% Sensitivity 78.20% Specificity 94.60% AUC 0.81
Di et al. [69]	V1–V3 transition index: [(SPVC/SSR)V1 + (SPVC/SSR)V2] − [(RPVC/RSR)V1 + (RPVC/RSR)V2 + (RPVC/RSR)V3] > −1.60	Sensitivity 93.00% Specificity 86.00% AUC 0.931
Efremidis et al. [70]	RV1-V3 transition ratio [(RV1 + RV2 + RV3)PVC/(RV1 + RV2 + RV3)SR] ≥ 0.9 predicts LVOT origin.	Sensitivity 94.00% Specificity 73.00% AUC 0.856
Zhao et al. [71]	R-S difference index (V2R + V3R +V4R − V1S) − of PVCs > 20.9 predicts LVOT origin.	Sensitivity 73.30% Specificity 86.30% AUC 0.867

### 11.2. Alternative ECG Configuration Algorithms (Table 3 and Table 4)

#### 11.2.1. Earliest Onset in V2 and Time to the First QRS Peak/Nadir in Leads V2 and III

##### QRS Morphology in V5R

Igarashi et al. [72] implemented the synthesized 18-lead ECG, where the right-chest leads (V3R, V4R and V5R) and the back leads (V7, V8, V9) are provided by mathematical computation, without the need to use additional leads or techniques. The authors concluded that QRS morphology in lead V5R could distinguish RVOT arrhythmias, which demonstrated an RS biphasic pattern, including Rs and rS morphology, from LVOT arrhythmias. In the prospective validation arm of 73 patients, V5R RS biphasic pattern predicted RVOT origin with 87% sensitivity and 91% specificity.

#### 11.2.2. R/S Concordance in Synthesized V3R, V4R and V5R

An additional synthesized right precordial lead study was conducted by Nakano et al. [73]. The authors sought to separate LVOT-derived arrhythmias from RVOT septal and RVOT free wall arrhythmias. LVOT arrhythmias exhibited in their entirety R > S concordance in synthesized right precordial leads (Syn-V3R, Syn-V4R and Syn-V5R), with 100% sensitivity and specificity. A transitional zone among synthesized right precordial leads was indicative of an RVOT septum origin (85% sensitivity, 100% specificity). Finally, RVOT free wall arrhythmias had R < S concordance, with 100% sensitivity and 85% specificity.

#### 11.2.3. V4/V8 Index

Zhang et al. [74] modified lead V5 to a posterior position, at the inferior tip of the left scapular, to create the lead V8. The V4/V8 index was defined as the ratio of PVC V4 R wave amplitude to V8 R wave amplitude, divided by the ratio of sinus rhythm V4 R wave amplitude to V8 R wave amplitude, i.e., (RPVCV4/RPVCV8)/(RSRV4/RSRV8). The authors hypothesized that PVCs originating from LVOT would exhibit a greater V4/V8 index because of the anterior-heading wave of the depolarization. A V4/V8 index cutoff of 2.28 yielded 67% sensitivity, 96% specificity, and a 89% positive predictive value for left side originating arrhythmias.

#### 11.2.4. V3R/V7 Index

Cheng et al. [75] implemented a V3R lead, placed at the right respective place of a V3 lead and V7 lead, placed to the left posterior axillary line of the fifth intercostal space, to create the V3R/V7 index. The V3R/V7 index was defined as the ratio of RPVCV3R amplitude to RPVCV7 amplitude. This novel index predicted LVOT origin with 87% sensitivity and 96% specificity, in the prospective validation cohort, when the threshold was set to ≥0.85. The authors reported that the rightward location of lead V3R, compared with the conventional leads, caused greater propagation from an LVOT focus. Respectively, an RVOT origin would result in a greater propagation in the leftward-located lead V7 [75].

**Table 3 diagnostics-13-03094-t003:** Additional lead algorithms–methodology.

Author	Year Published	Site Differentiation	Patients Included	Study Methodology	Electroanatomical Mapping	Inclusion Criteria
Igarashi et al. [72]	2014	RVOT vs. LVOT	101	Retrospertive cohort: 28 patients Prospective cohort: 73 patients	Yes	Idiopathic, symptomatic, drug-refractory ventricular arrhythmia Single bundle branch block with inferior axis on surface ECG
Nakano et al. [73]	2014	RVOT septum vs. RVOT free wall vs. LVOT	63	Retrospective	16 patients	Absence of obvious structural heart disease Successful RF ablation of symptomatic arrhythmia
Zhang et al. [74]	2017	RVOT vs. LVOT	174	Derivation Cohort: 134 patients Validation Cohort: 40 patients	Yes	Absence of structural heart disease, permanent pacing and bundle branch block
Cheng et al. [75]	2018	RVOT vs. LVOT	191	Derivation Cohort: 97 patiens Validation Cohort: 94 patients	Yes	Absence of coronary heart disease, structural heart disease, paced rhythm and preexisting bundle branch block during sinus rhythm

**Table 4 diagnostics-13-03094-t004:** Additional lead algorithms—diagnostic measures.

Author	Algorithm	Diagnostic Measures
Igarashi et al. [72]	RS biphasic pattern predicts RVOT morphology (both Rs and rS).	Sensitivity 87% Specificity 91%
Nakano et al. [73]	R > S concordance in all synthesized right precordial leads predicts LVOT origin. R < S concordance in all synthesized right precordial leads predicts RVOT-Free wall origin. transitional zone among synthesized right precordial leads predicts RVOT-septum origin	Sensitivity 100% Specificity 100% for Criterion A Sensitivity 100% Specificity 85% for Criterion B Sensitivity 85% Specificity 100% for Criterion C
Zhang et al. [74]	V4/V8 index: (R_PVC_V4/R_PVC_V8)/(R_SR_V4/R_SR_V8) > 2.28 predicts LVOT origin	Sensitivity 67% Specificity 96%
Cheng et al. [75]	R_PVC_V3R/R_PVC_V7 ≥ 0.85 predicts LVOT	Sensitivity 87% Specificity 96%

### 11.3. Artificial-Intelligence-Derived Algorithms (Table 5 and Table 6)

Recently, sophisticated machine learning methods have been utilized to differentiate the origins of outflow tract arrhythmias. We identified three studies that have used artificial intelligence to distinguish RVOT from LVOT arrhythmias.

#### 11.3.1. Gradient Boosting Method

Zheng et al. [76] applied an automated ECG feature extraction method to acquire 1,600,800 ECG features. The derived data were used as input for training an extreme gradient boosting tree algorithm. Extreme gradient boosting is a machine learning algorithm which combines multiple weak predictors to create a more robust model. The machine-learning-derived algorithm exhibited a sensitivity of 97% and a specificity of 100.0% in localizing RVOT arrhythmias and was superior to manual extraction of ECG data and previous diagnostic algorithms.

#### 11.3.2. Visualization Deep Learning Model

Nakasone et al. [77] implemented a deep learning model to diagnose the arrhythmia origins. Deep learning models constitute a subset of machine learning models and are composed of multiple layers of intervening nodes, also referred to as neurons [78]. Additionally, the authors applied gradient-weighted class activation mapping, to calculate the diagnostic contribution of each lead. Interestingly, arrhythmias originating from the right side of RVOT demonstrated larger QS wave in lead aVR, compared with lead aVL. Arrhythmias deriving from the left side of RVOT and LVOT exhibited larger QS wave in lead aVL, compared with lead aVR. The average prediction results were 95.2% for sensitivity and 92.0% for positive predictive value.

#### 11.3.3. Decision Tree Analysis

Shimojo et al. [79] developed a novel algorithm, using decision tree analysis machine learning, in a cohort of patients with outflow tract PVCs and precordial transition in lead V3. The parameters fed to the model included the R-wave amplitude ratio in lead aVF to lead II, the V2S/V3R index, the QRS amplitude in lead V3 and the ratio of R wave amplitude to the R wave deflection interval, in lead V3. The algorithm demonstrated 100% sensitivity and 91.5% positive predictive value for distinction of RVOT and LVOT arrhythmias.

**Table 5 diagnostics-13-03094-t005:** Machine-learning -derived algorithms—methodology.

Author	Year Published	Site Differentiation	Patients Included	Study Methodology	Electroanatomical Mapping	Inclusion Criteria
Zheng et al. [76]	2021	RVOT vs. LVOT	420	Training cohort: 340 patients Validation cohort: 38 patients Testing cohort: 42 patients	Yes	PVC or VT burden > 10% of total test duration
Nakasone et al. [77]	2022	RVOT vs. LVOT	80	Retrospective	Yes	NR
Shimojo et al. [79]	2023	RVOT vs. LVOT	104	Retrospective Training cohort: 72 patients Testing cohort: 32 patients	Yes	Precordial transition in lead V3 Left bundle branch block pattern Inferior axis QRS morphology

**Table 6 diagnostics-13-03094-t006:** Machine-learning-derived algorithms–diagnostic measures.

Author	Algorithm	Diagnostic Measures
Zheng et al. [76]	Extreme gradient boosting tree model, applied to automatically extracted ECG features	Sensitivity 96.97% Specificity 100%
Nakasone et al. [77]	Deep learning model, gradient—weighted class activation mapping method	Sensitivity 95.2% Positive Predictive Value 92.0%
Shimojo et al. [79]	Decision tree with maximum depth of three	Sensitivity 100% Positive Predictive Value 91.5%

## 12. Disadvantages of ECG Prediction Algorithms

An issue encountered is the retrospective and single-center nature of a proportion of the proposed algorithms, which limits their external validity. What is more, the complexity of several algorithms may limit their use in everyday clinical practice. The earlier algorithms exhibit diagnostic flaws due to cardiac rotation, although transition ratio and transition zone index mostly negate its effects [80]. An inherent limitation of all algorithms might also be the inaccurate identification of the side ablation eliminating PVCs, taking into account that there are cases with a deep myocardial origin and successful ablation on both sides of the myocardium.

## 13. Management of Idiopathic PVCs

Frequently, the primary role of the physician is to reassure the patient about the benign nature of the arrhythmia. The decision to treat idiopathic PVCs should be based upon the presence of symptoms or the development of PVC-induced myocardial dysfunction. Additionally, the clinician should take into consideration the PVC burden and PVC characteristics, which are associated with adverse prognosis, including epicardial [32], and interpolated [34] PVCs.

Therapeutic options include medical treatment and catheter ablation. Either beta blockers or nondihydropyridine calcium channel antagonists (CCBs) are perceived as first-line medical treatment options. In case of a higher burden of PVCs with higher heart rate or during exercise, beta-blockers should be preferred [81]. Beta blockers appear to be effective in symptom suppression, as they reduce post-extrasystolic potentiation via the Frank–Starling mechanism. In a recent observational study, beta blockers demonstrated effectiveness in reducing PVC burden in one third of the patients taking them [82]. Most importantly, if beta blockers do not suppress PVCs, they deteriorate symptoms due to the worsening of the bradysphygmia phenomenon.

CCBs are considered particularly efficient in fascicular PVCs [83]. Contemporary data indicate that class I-C antiarrhythmic drugs, namely flecainide and propafenone, can effectively suppress idiopathic PVCs and lead to EF recovery, among patients with PVC-induced myocardial dysfunction [84]. These findings highlight the crucial role of substrate regarding medication effects on the myocardium (IC antiarrhythmics fared well in PVC-induced myocardial dysfunction—no abnormal substrate is present—and much worse in post-infarction, non-revascularized myocardia). Amiodarone has a role in PVC suppression only in cases of structural heart disease due to the associated side effect profile, especially upon long term administration [85,86].

Catheter ablation is an effective option to reduce or eliminate PVCs. In a multicenter cohort study of 1185 patients, acute procedural success was accomplished in 84% of the participants [35]. Catheter ablation demonstrates superior efficacy and safety compared with anti-arrhythmic drug therapy, especially in RVOT-originating arrhythmias [87]. These results are consistent with a recent meta-analysis among randomized and non-randomized trials [88]. Catheter ablation was associated with reduced PVC recurrence, frequency, and burden. Complication and adverse event rates were lower in the catheter ablation group. The majority of complications from catheter ablation are related to vascular access and include pseudoaneurysm, hematoma, and arteriovenous fistula. Major complications are rare and include mostly pericardial tamponade and atrioventricular block [35].

## 14. Conclusions

PVCs are a frequent finding among the general population. The prognosis and natural history of PVCs is still a matter of debate and PVC patients are, under certain conditions, more prone to developing PVC-induced myocardial dysfunction. The outflow tract constitutes the most common site of origin of PVCs, with RVOT dominance over LVOT, especially in younger groups. Each outflow tract site produces a distinct ECG PVC pattern, which can be used for the localization of the arrhythmia origin site. Numerous algorithms have been developed to capitalize on the distinct ECG morphologies for the purpose of separating the origin site between RVOT and LVOT. Novel algorithms include alternative ECG-lead configurations and implement artificial intelligence. Ultimately, the decision to treat PVCs is based upon the presence of symptoms or the development of myocardial dysfunction. Treatment options include antiarrhythmic drugs and catheter ablation. Further research is required to elucidate the precise natural history and prognosis, to effectively distinguish RVOT from LVOT arrhythmia origins and to define the most appropriate treatment modalities.

## Figures and Tables

**Figure 1 diagnostics-13-03094-f001:**
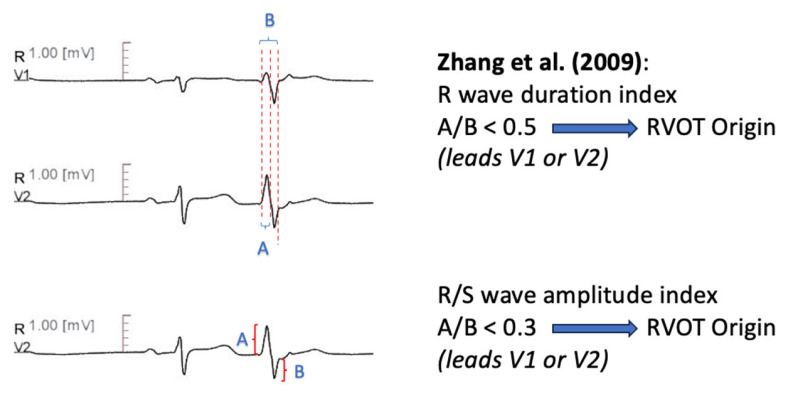
Graphical representation of R Wave Duration Index- R/S Amplitude Index [60].

**Figure 2 diagnostics-13-03094-f002:**
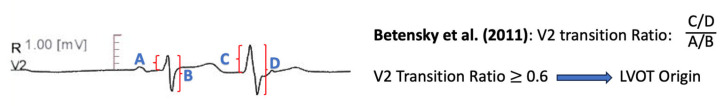
Graphical representation of V2 Transition Ratio [61].

**Figure 3 diagnostics-13-03094-f003:**
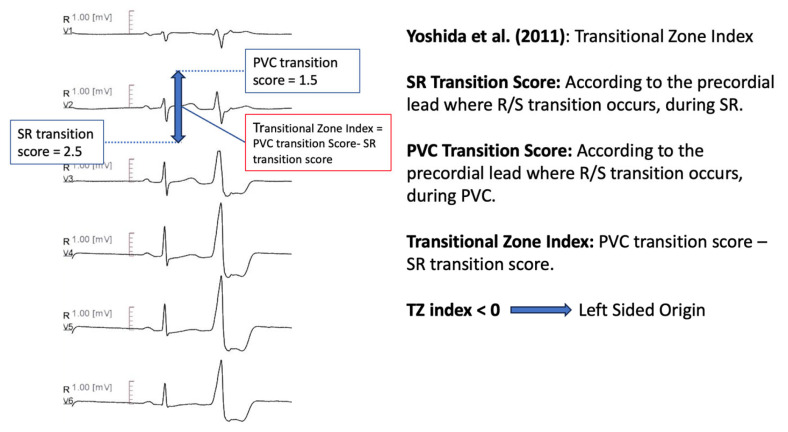
Graphical representation of Transitional Zone Index [62].

**Figure 4 diagnostics-13-03094-f004:**
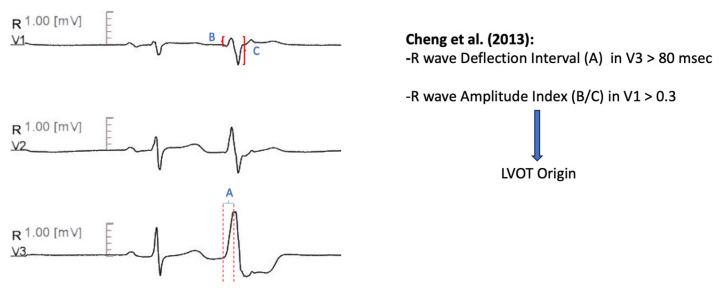
Graphical representation of V3 R-Wave Deflection Interval combined with R-Wave Amplitude Index [63].

**Figure 5 diagnostics-13-03094-f005:**
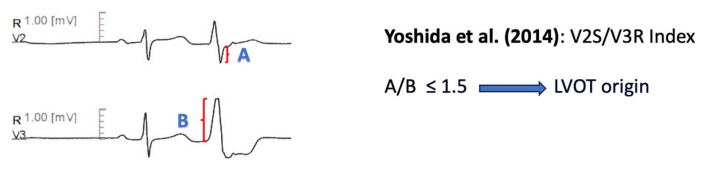
Graphical representation of V2S/V3R Amplitude Index [64].

**Figure 6 diagnostics-13-03094-f006:**
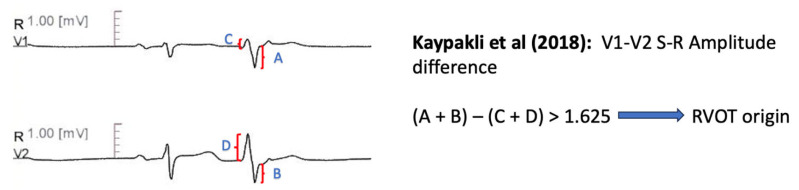
Graphical representation of V1–V2 S-R Amplitude Difference [65].

**Figure 7 diagnostics-13-03094-f007:**
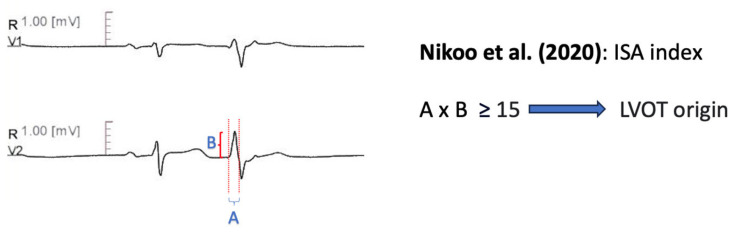
Graphical representation of Initial R Wave Surface Area Index [68].

**Figure 8 diagnostics-13-03094-f008:**
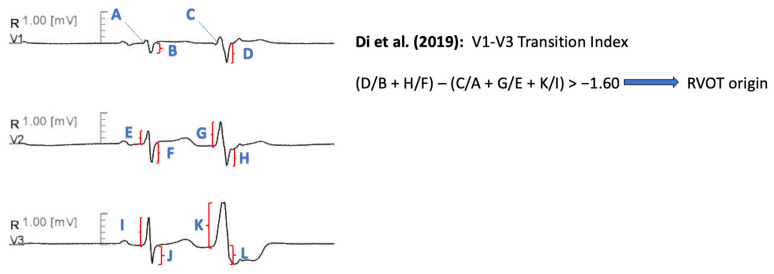
Graphical representation of V1–V3 Transition Index [69].

**Figure 9 diagnostics-13-03094-f009:**
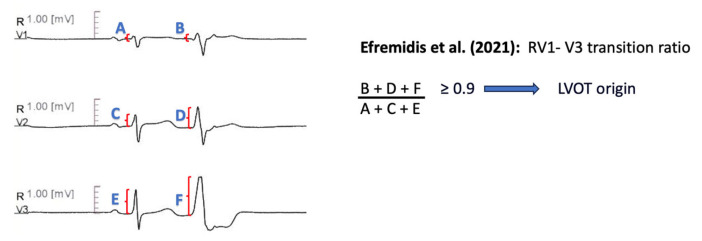
Graphical representation of RV1-V3 Transition Ratio [70].

**Figure 10 diagnostics-13-03094-f010:**
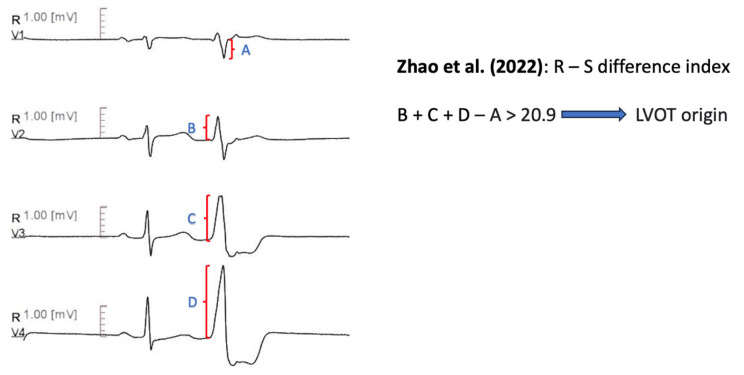
Graphical representation of R-S Difference Index [71].

## Data Availability

No new data were generated during this study.

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
