# Peer review of "Electrocardiographic Characteristics, Identification, and Management of Frequent Premature Ventricular Contractions"

_diagnostics, 2023, doi:10.3390/diagnostics13193094_

Round 1

Reviewer 1 Report

General comments:

This is a review about the association of PVCs with CV outcomes in the absence of structural heart disease. The authors review the recent literature for the prognostic value of CMR and PVC burden as well as the value of ECG and AI for localizing PVC-origin. Finally, the focus on symptoms and myocardial dysfunction to guide therapy with BB, I-C antiarrhythmic drugs or ablation, emphasizing the efficacy and safety of the later.

The manuscript is well written and the conclusions seem appropriate. The summary of different algorithms for the PVC origin has practical value. Some minor issues should be addressed prior to considering publication in the Journal of Diagnostics.

Specific comments:

1.      Prognosis: The authors report that “However, the suppression of PVCs in survivors of myocardial infarction led to detrimental effects on total mortality, challenging the causal relationship between PVCs and adverse outcomes in ischemic heart disease [16].” This is based on the CAST trial and refers only to flecainide, a very commonly used drug at the time, that was proven to be detrimental for ischemic patients. This is now known and flecainide is contraindicated in those patients. However, this alone does not challenge the causal relationship between PVCs and adverse outcomes in ischemic heart disease. Please consider rephrasing and supplementing with data that relate PVCs with prognosis in ischemic patients with or without appropriate antiarrhythmic drugs.

2.      Disadvantages of ECG Prediction Algorithms: Most studies identify the side of the PVC origin according to the side ablation eliminated or reduced the PVCs. However, some PVCs have a deeper myocardial origin (e.g. from the LV summit or aortomitral continuity) and ablation on both sides of the myocardium is applied for success. Most studies do not report these cases separately and thus do not adjust their results accordingly (bias). Consider commenting

3.      Classic ECG Algorithms: Consider providing an example with calculation of the different indexes in order to increase readability and practical use for the reader.

Author Response

We would like to thank the reviewer for the kind effort reviewing our manuscript. 

  1. We totally agree that CAST results cannot change the causal relationship between PVCs and adverse outcomes in ischemic heart disease, as noticed in references 13-15. We have rephrased the relative comment as required.
  2.  We also totally agree that an inherent limitation of all algorithms might also be the inaccurate identification of the side ablation eliminating PVCs, taking into account that there are cases with a deep myocardial origin and successful ablation on both sides of the myocardium. This exact phrase has been added in the revised manuscript (section 12).
  3. We have added detailed schematic figures of most algorithms in order to increase readability and practical use for the reader.

---------------------------------------------------------------------------

Reviewer 2 Report

1.      The title and abstract fit the article correctly.

2.      The references are precise but can be better up-to-date, some even from the 70s and 80s.

3.      Enough scientific soundness.

4.    The abstract is clear and adequately synthesized.

 The English style is not uniform.  Some are American, while some are British.

The overall format, punctuation, and writing tone can be better (e.g., using points, decimals, and commas).

-          Double check lines 33, 54, 228, 234-236, 243, 254-255, 260, 266, 298, 306, 318, 323, 339, 353, 364-365, 372, etc.

PS.  Furthermore, some numbers in this article are weird, like lines 30 and 350.  Would you please explain them more?

Author Response

We would like to thank the reviewer for the kind comments.

We have performed a thorough revision of the revised manuscript regarding the English style as depicted throughout the whole manuscript.

We have also corrected the overall format, punctuation, and writing tone as well as corrected weird numbers in specific parts.

---------------------------------------------------------------------------

Reviewer 3 Report

The largest part of the article concerns the electrocardiographic analysis of the morphology of ventricular extrasystoles depending on the location of the arrhythmia.

The authors, based on well-selected literature, present the most common ECG patterns of PVCs.

Including examples of ECG records in the article would significantly enrich the work and make it more clinically useful.

Author Response

We would like to thank the reviewer for the kind comments after reviewing our manuscript. We have includied examples of ECG records in the article in order to make it more clinically useful according to your suggestion.

---------------------------------------------------------------------------

Reviewer 4 Report

Dear authors,

The first part of the manuscript seems to be written by AI programs. 

I suggest to rephrase them.

Although the main focus of this paper is ECG and echocardiography, we did not add any images of ECG and echo for a better understanding.

Author Response

We would like to thank the reviewer for reviewing our manuscript.

We have included examples of ECG records in the article in order to make it more clinically useful according to your suggestion.

We have also performed a thorough revision of the revised manuscript regarding the English style as depicted throughout the whole manuscript.

====================================

Round 2

Reviewer 4 Report

Congratulations! 

The authors have revised accordingly.

I agree with the publication in its present form.